

# Construction of a *fur* null mutant and RNA-sequencing provide deeper global understanding of the *Aliivibrio salmonicida* Fur regulon

Sunniva Katharina Thode, Cecilie Bækkedal, Jenny Johansson Söderberg, Erik Hjerde, Hilde Hansen and Peik Haugen

Department of Chemistry and The Norwegian Structural Biology Centre, Faculty of Science and Technology, UiT The Arctic University of Norway, Tromsø, Norway

## ABSTRACT

**Background**. The ferric uptake regulator (Fur) is a transcription factor and the main regulator of iron acquisition in prokaryotes. When bound to ferric iron, Fur recognizes its DNA binding site and generally executes its function by repressing transcription of its target genes. Due to its importance in virulence, the Fur regulon is well studied for several model bacteria. In our previous work, we used computational predictions and microarray to gain insights into Fur-regulation in *Aliivibrio salmonicida*, and have identified a number of genes and operons that appear to be under direct control of Fur. To provide a more accurate and deeper global understanding of the biological role of Fur we have now generated an *A. salmonicida fur* knock-out strain and used RNA-sequencing to compare gene expression between the wild-type and *fur* null mutant strains.

**Results**. An *A. salmonicida fur* null mutant strain was constructed. Biological assays demonstrate that deletion of *fur* results in loss of fitness, with reduced growth rates, and reduced abilities to withstand low-iron conditions, and oxidative stress. When comparing expression levels in the wild-type and the *fur* null mutant we retrieved 296 differentially expressed genes distributed among 18 of 21 functional classes of genes. A gene cluster encoding biosynthesis of the siderophore bisucaberin represented the highest up-regulated genes in the *fur* null mutant. Other highly up-regulated genes all encode proteins important for iron acquisition. Potential targets for the RyhB sRNA was predicted from the list of down-regulated genes, and significant complementarities were found between RyhB and mRNAs of the *fur*, *sodB*, *cysN* and VSAL_I0422 genes. Other sRNAs with potential functions in iron homeostasis were identified.

**Conclusion**. The present work provides by far the most comprehensive and deepest understanding of the Fur regulon in *A. salmonicida* to date. Our data also contribute to a better understanding of how Fur plays a key role in iron homeostasis in bacteria in general, and help to show how Fur orchestrates iron uptake when iron levels are extremely low.

Corresponding author
Peik Haugen, peik.haugen@uit.no

# INTRODUCTION

The ferric uptake regulator, Fur, represents the main regulator of iron levels in prokaryotic microorganisms (reviewed by *Fillat, 2014*). In addition to regulating iron acquisition genes, Fur also regulates genes involved in e.g., the TCA cycle, DNA metabolism, energy metabolism, redox-stress resistance, chemotaxis, swarming, metabolic pathways, toxin production and other virulence factors, and is therefore considered as a so-called master regulator (*Escolar, Perez-Martin & Lorenzo, 1999*; *Hantke, 2001*; *McHugh et al., 2003*; *Mey et al., 2005*; *Pajuelo et al., 2016*). Transcriptomic studies on *fur* null mutants of *Vibrio cholerae* (*Mey et al., 2005*) and *Vibrio vulnificus* (*Pajuelo et al., 2016*) have shown that Fur represses expression of siderophore biosynthesis and transport genes, heme transport and utilization genes, ferric and ferrous iron transport genes, stress response and biofilm genes, amongst others. The same studies also showed that Fur activates genes involved in stress responses, chemotaxis, motility and toxin production. In *Escherichia coli* K-12, Fur directly regulates 131 genes including those of seven other master regulators, i.e., *flhD, flhC, felc, soxS, ryhB, rpoS* and *purR* (*McHugh et al., 2003*), which subsequently can result in regulation of 3158 genes in total (incl. direct and indirect effects), according to EcoCyc (*Keseler et al., 2013*). This huge number of genes translates to >70% of the total number of genes in *E. coli* K-12 (which is 4318 according to EcoCyc), and illustrates the central role of Fur in cellular processes far beyond iron homeostasis.

The 3D-structure of Fur from *Pseudomonas aeruginosa*, *E. coli*, *V. cholerae*, *Helicobacter pyroli* and *Campylobacter jejuni* is known (*Butcher et al., 2012*; *Dian et al., 2011*; *Pecqueur et al., 2006*; *Pohl et al., 2003*; *Sheikh & Taylor, 2009*). These structures show that Fur mainly acts as a homodimer in both apo and holo forms, where at least two zinc ligands per monomer stabilize the dimer (*Fillat, 2014*). The iron binding sites are located in a DNA binding domain of each monomer. Here, iron binding causes conformational changes that enable Fur to bind to its DNA target (known as the Fur-box) (*Fillat, 2014*). Although several different Fur-box motifs have been proposed over the years, the current literature have converged on a 19 bp palindromic sequence centered around a non-conserved position (*Ahmad et al., 2009*; *Baichoo & Helmann, 2002*; *Davies, Bogard & Mekalanos, 2011*; *De Lorenzo et al., 1988*; *Escolar, Perez-Martin & Lorenzo, 1998*). Once bound to its DNA target, Fur mainly acts as a repressive regulator by blocking the transcription of downstream genes.

An apparent gene activating effect by Fur was observed during early investigations of the Fur regulon and was proposed to be due to post-transcriptional regulation (*Hantke, 2001*). This effect was later discovered to originate from negative regulation by Fur of a gene encoding the small regulatory RNA (sRNA) named RyhB (*Masse, Escorcia & Gottesman, 2003*; *Massé & Gottesman, 2002*; *Masse, Vanderpool & Gottesman, 2005*). The RyhB sRNA is responsible for destabilizing mRNAs of its target genes, and repression of *ryhB* by holo-Fur was therefore interpreted as activation by Fur. RyhB typically targets mRNAs encoding iron-using or iron-binding proteins as a way of preserving the iron levels in the cell at low iron conditions (*Davis et al., 2005*; *Masse, Vanderpool & Gottesman, 2005*; *Murphy & Payne, 2007*). In *E. coli* RyhB directly targets 28 mRNAs (see http://ecocyc.org/). Examples of targets include mRNAs of *bfr, cysE, sodAB, fumA, sucBCD, icsRSUA,* and *sdhABCD*
(*Massé & Gottesman, 2002*). In *V. cholerae* RyhB targets mRNAs of *sodB*, *sdhC*, *gltB1* and *fumA*. In contrast to *E. coli,* mRNAs of the iron storage genes like *bfr* and *ftn* are not regulated by the *V. cholerae* RyhB (*Davis et al., 2005*).

The aim of this study was to investigate the Fur regulon in *A. salmonicida*, the causative agent of cold-water vibriosis in Atlantic salmon (*Salmo salar*), rainbow trout (*Oncorhynchus mykiss*) and Atlantic cod (*Gadus morhua*) at sea-water temperatures below 10 °C (*Colquhoun & Sorum, 2001*; *Enger, Husevag & Goksoyr, 1991*). In a previous study we identified a *Vibrionaceae*-specific Fur-box consensus as 5′-AATGANAATNATTNTCATT-3′, and used computational methods to predict Fur-regulated genes and operons in four *Vibrionaceae* genomes, including *A. salmonicida* (*Ahmad et al., 2009*). Fur-binding motifs were associated with 60 single genes and 20 operons (89 genes in total). Later we used molecular dynamics (MD) simulations and binding free energy calculations to gain more insights into the interactions between *A. salmonicida* Fur (asFur) and proposed Fur-binding sites (*Pedersen et al., 2010*). Here, Fur-binding to promoters was dependent on the number of Fur-boxes, and the predicted "strengths" (i.e., calculated similarity to Fur-box consensus) of the individual Fur-boxes. Finally, we studied Fur-regulation in *A. salmonicida* using iron-depletion experiments in combination with custom whole-genome microarray chips (*Ahmad et al., 2012*; *Thode et al., 2015*). Thirty-two genes were found to be significantly up-regulated 15 min after exposure to low-iron conditions (suggesting Fur-regulation), and interestingly, the *bibABC* genes encoding the producing proteins for the siderophore bisucaberin were identified as being most highly up-regulated (*Thode et al., 2015*). We have now constructed an *A. salmonicida fur* null mutant and used Illumina based RNA-sequencing (RNA-seq) to compare the transcriptomes of the wild-type strain and the *fur* null mutant. Overall, we find that the RNA-seq data overlap remarkably well with our previous findings when using microarray. However, we also show that high-throughput RNA-sequencing provide us with a much more accurate and fine-grained global understanding of the Fur regulon in *A. salmonicida,* compared to what we knew from our previous microarray work.

## MATERIALS AND METHODS

### Bacterial strains, culture conditions, and sampling for RNA sequencing

*A. salmonicida* LFI1238 (*Hjerde et al., 2008*) was used as parental strain for the construction of the *A. salmonicida fur* null mutant (see below for details). Parental and mutant strains were cultured in LB medium (Luria-Bertani broth Miller, Difco (later corrected to Lysogeny Broth (*Bertani, 2004*))) containing 2.5% NaCl at 12 °C and 200 rpm. For *E. coli* strain S17-1 the growth conditions were 37 °C and 200 rpm in LB medium with 1% NaCl. The suicide plasmid pDM4 (*Milton et al., 1996*) was propagated in *E. coli* S17-1 cells. For selection of *E. coli* S17-1 transformants and *A. salmonicida* transconjugants, 25 μg or 2 μg of chloramphenicol/ml was added to the medium, respectively.

For biological characterizations (see below for details) and RNA sequencing sampling, *A. salmonicida* LFI1238 and *fur* null mutant strains were cultured in LB medium with 1%

NaCl at 8 °C and 200 rpm. For RNA sequencing, three biological replicates of *A. salmonicida* LFI1238 and *A.salmonicida fur* null mutant were grown to mid log growth phase, i.e., at optical density (600 nm) of approximately 0.5. Ten mL samples were harvested, spun down and the cell pellets were then stored at −80 °C for later processing.

## Construction of an *A. salmonicida fur* null mutant

The *A. salmonicida fur* null mutant was constructed using the suicidal plasmid pDM4 (a map of pDM4 can be found at https://www.google.com/patents/EP1425037B1?cl=en) and allelic exchange, as described by others (*Milton et al., 1996*). First we constructed the plasmid pDM4Δ*fur*, consisting of merged flanking regions of the *fur* gene. The upstream flanking region of the *fur* gene was amplified by PCR using primers FurA forward (5′-CTACTCGAGATATTTATTTCCCTTTAATTC-3′) and FurB reverse (5′-CACGTAAACTAAATATGACTTTTCCTGTATTGG-3′). For amplification of the down-stream flanking region, primers FurC forward (5′-TATTTAGTTTACGTGCATAAAAAA-3′) and FurD reverse (5′-CCCACTAGTATAACAAAGACTCTACTCCAG-3′) were used. The resulting upstream and downstream PCR products were fused together using an overlap PCR, cut with restriction enzymes *XhoI* and *SpeI*, and ligated into the corresponding sites of pDM4. The resulting pDM4Δ*fur* construct was transformed into *E.coli* S17-1 and used as donor cells in conjugation experiments with *A. salmonicida* as described elsewhere (*Bjelland et al., 2012*). Briefly, *E. coli* S-17 transformed with pDM4Δ*fur* was cultivated to mid-log phase and *A. salmonicida* LFI1238 to stationary phase before they were harvested, centrifuged, and washed with LB containing 1% NaCl. Donor and recipient strains were resuspended and spottet on LB agar containing 1% NaCl and incubated at room temperature for 6 h to stimulate conjugation, then at 12 °C for 15 h to provide better growth conditions for *A. salmonicida*. Spotted cells were suspended in LB containing 2.5% NaCl and incubated at 12 °C with 200 rpm for 24 h. Next, cultures were spread on LB agar containing 2.5% NaCl and 2 μg/ml CAM and incubated at 12 °C for four days. Potential transconjugants were verified using PCR. Transconjugants were spread on LB agar containing 5% sucrose to promote allelic exchange. Disposition of pDM4 was verified using a CAM sensitivity test and *A. salmonicida fur* null mutant was verified using PCR (see Fig. S1A) and DNA sequencing (see Fig. S1B) with primers FurE (5′-ATTGGGTACGATTCGCATTC-3′) and FurF (5′-TTCACAGTGCCAAACTCTGC-3′).

## Total RNA purifications

For RNA-seq, total RNA was purified from cell pellets using the Masterpure complete DNA & RNA purification kit (Epicentre, Madison, WI, USA) following the manufacturer's protocol, followed by an additional DNA removal step using the DNA-free kit (Applied Biosystems, Foster City, CA, USA). DNase-treated total RNA was subsequently purified using the RNA cleanup RNeasy MinElute kit (Quigen, Hilden, Germany). The quality of total RNA preps was determined using a Bioanalyzer and a Prokaryote Total RNA Pico Chip (Agilent Technologies, Foster City, CA, USA). Finally, ribosomal (r) RNA was removed from each sample (5 μg total RNA) using the Ribo-Zero rRNA Removal Kit (bacteria) (Epicentre, Madison, WI, USA) according to the manufacturer's instructions.

rRNA-depleted RNA samples were ethanol precipitated (to recover small RNAs), and analyzed on a Bioanalyzer using mRNA Pico Chips (Agilent Technologies, Santa Clara, CA, USA).

## RNA-sequencing and data analysis

RNA-seq libraries were generated from purified rRNA-depleted RNA samples using the strand-specific TruSeq stranded mRNA library prep kit (Illumina, San Diego, CA, USA), and sequenced at the Norwegian Sequencing Centre using the Illumina NextSeq 500 with mid output reagents with a read length of 75 bp and paired end reads. Details on the RNA-seq data is provided in Table S1. The reads were quality checked using FastQC. Further analysis of the RNA-Seq data was performed using a Galaxy pipeline consisting of EDGE-pro v1.0.1 (Estimated Degree of Gene Expression in Prokaryotes) and DESeq. EDGE-pro was used to align the reads to the *A. salmonicida* LFI1238 genome (*Hjerde et al., 2008*), and to estimate gene expression. Differences in gene expression between wild-type and *fur* null mutant were determined using DESeq. Log$_2$ fold changes of the genes were recalculated to $\times$ differential expression values (i.e., $\Delta fur$/wt) and genes were defined as significantly differentially expressed based on a $p$-value $\leq 0.05$ and differentially expression values of $\Delta fur$/wt $\geq 2\times$ and $\leq -2\times$.

## sRNA and mRNA target predictions

The Rockhopper software (*McClure et al., 2013*) was used to identify sRNA from the RNA-seq data. Input files in the analysis were fastq files from the RNA-seq data, a protein coding gene position file (.ptt), a non-coding RNA position file (.rnt), and finally genome files from *A. salmonicida* LFI1238 (NC_011312.1 (Chr I), NC_011313.1 (ChrII), NC_011311.1 (pVSAL840), NC_011314.1 (pVSAL320), NC_011315.1 (pVSAL54) and NC_011316.1 (pVSAL43)). sRNAs identified by Rockhopper were visualized in Artemis and manually curated based on a set of criteria. To be accepted as a potential sRNA, its gene should be (i) located in an intergenic region, (ii) between 30–350 nt in length, (iii) located 30 nt or more from the nearest CDS if on the same strand, and 10 nt if on the complementary strand (based on the method of *Toffano-Nioche et al., 2012*). RNAs fulfilling the criteria described above were further examined for presence of small open reading frames (sORF) using a method adopted from *Van der Meulen, De Jong & Kok (2016)*, since there is an increasing awareness of their presence in bacterial genomes although their significance is not fully understood (*Hobbs et al., 2011*). Finally, EDGE-pro and DESeq was used to estimate differential gene expression levels for the sRNAs/sORFs.

TargetRNA2 and IntaRNA were used to identify potential sRNAs targets (*Busch, Richter & Backofen, 2008*; *Kery et al., 2014*). Using sRNA sequences as queries, the programs searches for complementary regions in 5′ regions of mRNAs. Only targets predicted by both programs were accepted. We also searched for mRNA targets for up-regulated sRNAs (ten sRNAs with folds $\Delta fur$/wt $\geq 2\times$ in the RNA-seq dataset), including RyhB, among the 34 most down-regulated genes in our RNA-seq data set. This was done to identify sRNAs with critical roles in iron homeostasis (similar to RyhB). In addition, we predicted binding between RyhB and its verified targets (*sodB*, *gltB*, *sdhC* and *fumA*) verified experimentally in

*E. coli* and *V. cholerae*. Nucleotide sequences of RyhB targets were extracted from European Nucleotide Archive (ENA). The nucleotide sequences were aligned with corresponding sequences in *A. salmonicida* and examined using Jalview (*Waterhouse et al., 2009*).

**Biological characterization of *A. salmonicida fur* null mutant**

*A. salmonicida* LFI1238 wt and *fur* null mutant ($\Delta fur$) were cultured in LB (Difco) at 8 °C and 200 rpm in all experiments. Growth of cultures was monitored with optical density measured at 600 nm. To determine growth effects of *fur* null mutation, four replicates of *A. salmonicida* LF1238 wt and $\Delta fur$ were cultured from lag phase until stationary phase. To determine the ability of the *fur* null mutant to withstand low iron conditions, wt and $\Delta fur$ cultures were first grown to $OD_{600\,nm}$ of 0.38 and 0.33 (mid log phase), respectively. The cultures were then split into five separate flasks. One culture was kept as control whereas 25–500 $\mu$M of the iron chelator 2, 2′-dipyridyl was added to the remaining cultures. To determine the ability of the *fur* null mutant to withstand oxidative conditions, wt and $\Delta fur$ cultures were first grown $OD_{600\,nm}$ of 0.4 and 0.35 (mid log phase), respectively. The cultures were then split into five separate flasks. One culture was kept as control whereas 50–1,000 $\mu$M of hydrogen peroxide was added to the remaining cultures. Growth was monitored for approximately 40 h.

## RESULTS AND DISCUSSION

**Construction and basic characterization of an *A. salmonicida fur* null mutant**

To better understand the Fur regulon in *A. salmonicida*, a *fur* null mutant was constructed using the genetic system described by *Milton et al. (1996)*. Briefly, approximately 250 bp of upstream and 250 bp downstream sequences flanking the *fur* gene were merged and inserted into the pDM4 suicide vector (contains *sacBR*), which was then transformed into *E. coli* S17-1 cells, and finally conjugated into *A. salmonicida* LFI1238 to trigger recombination and deletion of *fur*. The *fur* null mutant was verified by PCR and sequencing.

Basic characterization of the *fur* null mutant was done to examine the physiological and morphological effects of the *fur* deletion. Because Fur is a global regulator, we expected the *fur* null mutant to loose fitness due to loss of control of central cellular processes. For example, loss of Fur is expected to reduce the growth rate, and result in reduced ability to respond to external chemical stress, such as presence of $H_2O_2$ and iron chelators (*Becerra et al., 2014*; *Fillat, 2014*; *Hassett et al., 1996*; *Touati, 2000*; *Yang et al., 2013*). Effects on growth was monitored by comparing the growth rates of the wild-type and the *fur* null mutant in LB with 1% NaCl at 8 °C and 200 rpm shaking. The $OD_{600\,nm}$ of the starting cultures were set to 0.01 and then monitored until cultures reached stationary phase (typically $OD_{600\,nm}$ 1.2–1.4). The lag phase for the wt and *fur* null mutant lasted approximately 10 and 35 h, respectively, and doubling times were approximately 6 and 12 h during mid log phase (Figs. S2A and S2B). To test the ability to respond to chemical stress the *fur* null mutant and the wild-type strain were exposed to increasing concentrations of hydrogen peroxide ($H_2O_2$) and the iron chelator 2, 2′-dipyridyl. The minimum inhibitory concentration of $H_2O_2$ on growth for the wild-type and *fur* null mutant were 500 $\mu$M and 50 $\mu$M,

respectively (Figs. S3A and S3B). In a similar experimental setup with 2, 2′-dipyridyl the effects were less dramatic (Figs. S3C and S3D). The minimum inhibitory of 2, 2′-dipyridyl concentrations were similar (approx. 100 μM) for both wild-type and mutant strain. However, whereas the wild-type strain grows well in the presence of 1 mM 2, 2′-dipyridyl, the *fur* null mutant cannot grow in the presence of 500 μM.

In summary, deletion of the *fur* gene results in longer lag phase during growth, longer cell doubling time and reduced ability to respond to oxidative reagents and iron chelators. This is in agreement with results from other γ-proteobacteria model organisms, e.g., *V. vulnificus* Δ*fur* shows higher sensitivity to oxidative stress, reduced fitness and growth (*Pajuelo et al., 2016*) and *V. cholerae* Δ*fur* shows reduction in logarithmic growth (*Mey et al., 2005*), and support the validity of the *A. salmonicida fur* mutant.

## RNA-sequencing identifies 296 differentially expressed genes in the *A. salmonicida fur* null mutant

To provide accurate data on the Fur regulon we next compared the transcriptome of the *A. salmonicida fur* null mutant and the wild-type using an RNA-seq approach. RNA samples (from three biological replicates) were prepared from *A. salmonicida* LFI1238 wild-type and *fur* null mutant cells grown in LB containing 1% NaCl at 8 °C to mid log phase ($OD_{600\ nm} \approx 0.5$). The given temperature and salt concentration were chosen because *A. salmonicida* is responsible for development of cold-water vibriosis in Atlantic salmon (i.e., at physiological salt conditions) at temperatures below 10 °C (*Bergheim et al., 1990*; *Colquhoun & Sorum, 2001*). RNA samples from biological replicates were subjected separately to paired-end RNA-seq using Illumina NextSeq 500 with 75 bp read length. Sequencing generated an average output of approximately 54 million reads per sample. RNA-seq data was analyzed using a Galaxy pipeline running EDGE-pro v1.0.1 and DESeq. EDGE-pro was used to align reads to the *A. salmonicida* LFI1238 genome, and estimate gene expression. Comparison of gene expression between wild-type and *fur* null mutant was done using DESeq. Reads originating from rRNA and tRNA genes were excluded from the data analysis. Threshold values for differential expression were set to ≥2× difference (equal to $Log_2 = 1$), and with *p*-value ≤0.05.

Figure 1 shows how a total of 296 differentially expressed genes are distributed among functional gene classes (functional classes adapted from MultiFun (*Serres & Riley, 2000*)). One hundred sixty-two and 134 genes are up-regulated and down-regulated, respectively. The complete list of the 296 differentially expressed genes are presented in Table S2. All functional classes, except ''ribosome constituents'', ''nucleotide biosynthesis'' and ''cell division'', are represented, and the two classes ''cell envelope'' and ''transport/binding proteins'' contain the highest number of genes. Considerable up-regulation of genes from the two latter classes is expected since Fur generally regulates genes as a repressor (*Fillat, 2014*), and loss of Fur is therefore expected to result in up-regulation (in *fur* null mutant) of genes involved in iron binding and transport over the membranes. Down-regulated genes are more evenly distributed among 18 of the 21 functional classes, including central processes such as ''energy metabolism'', ''central metabolism'', ''amino acid biosynthesis'' and ''cell processes''. Although there is no clear pattern, the combined data of up-regulated

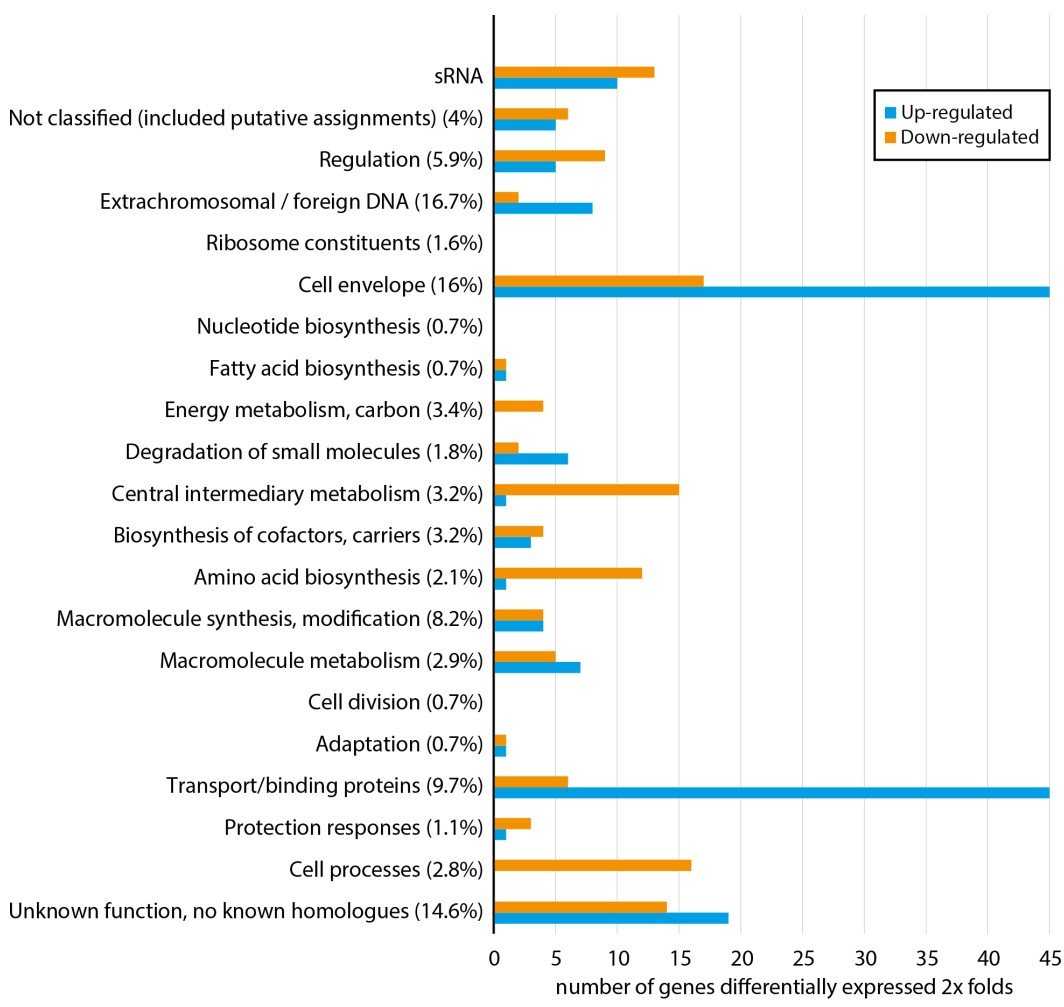

**Figure 1** **Functional distribution of genes that are ≥2× differentially expressed between *A. salmonicida* wild-type and a *fur* null mutant strain.** Numbers in parentheses represent percentage of the total number of genes within the genome in each functional class. For complete list of differentially expressed genes, see Table S2.

and down-regulated genes support that *as*Fur is a master regulator with functions similar to that of Fur in *E. coli* (*ec*Fur) (*McHugh et al., 2003*).

## Chromosomal distribution of differentially expressed genes

Tables 1 and 2 summarize details of genes and operons that are up- or down-regulated, Fig. 2 shows the chromosomal distribution and positions of the differentially expressed genes, and Fig. 3 shows details on RNA-seq reads mapped against the genome for a selection of genes and operons (that will be discussed in more detail below). Previous studies have shown a strong correlation between the distance of genes from *oriC* (Chr I), and their general transcription level (also known as the *gene dosage effect*) (*Dryselius et al., 2008*; *Toffano-Nioche et al., 2012*). That is, genes located close to *oriC* are, statistically, more likely to be transcribed at higher levels than genes located further away from *oriC*, and we

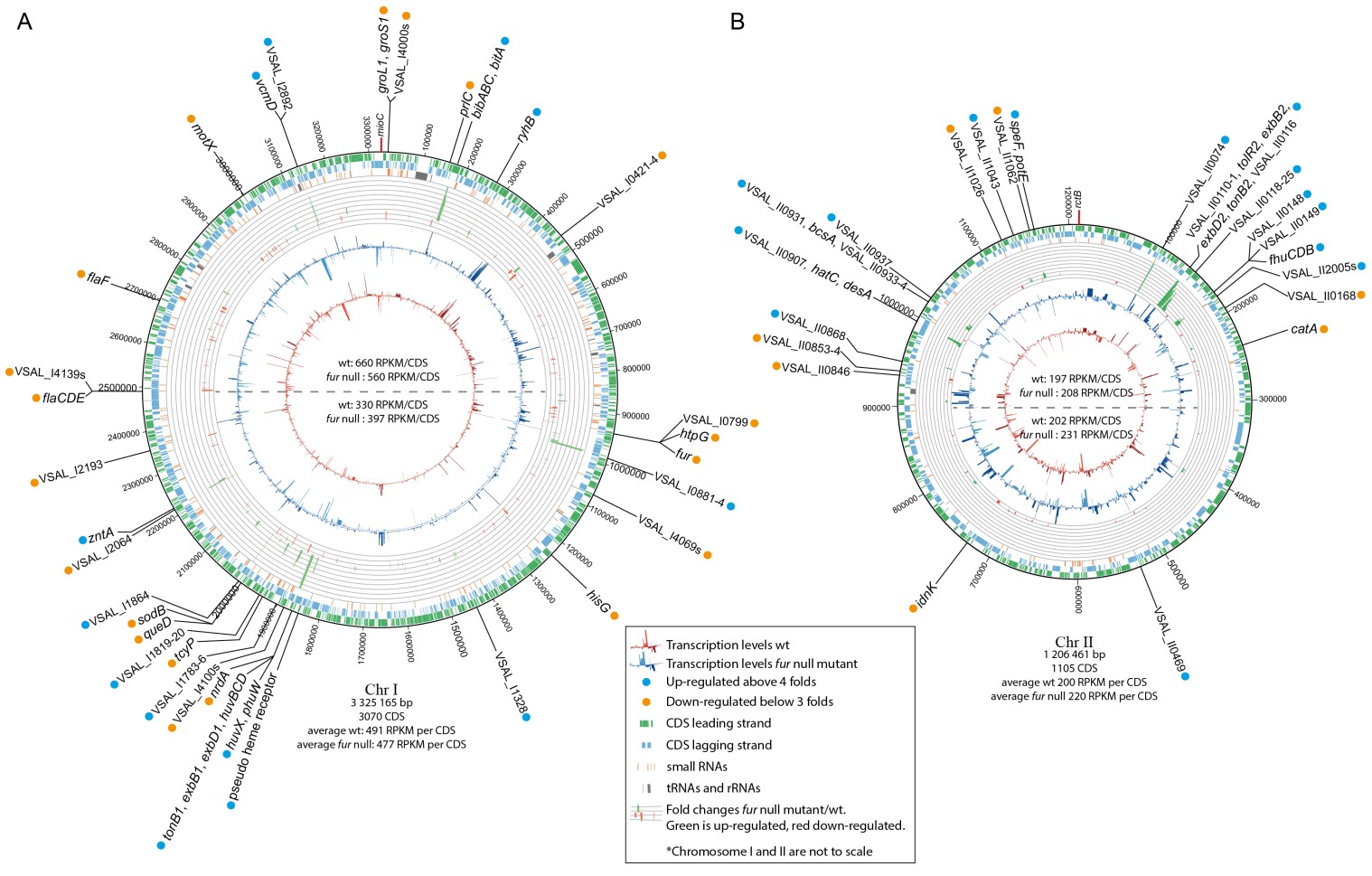

**Figure 2   Schematic circular diagrams of the *A. salmonicida* chromosomes I (A) and II (B) (ChrI and ChrII).**  The scale of the circles is in base-pairs. More than or equal to 4× differentially expressed genes are indicated with light blue filled circles and ≤ −3× differentially expressed genes are indicated with orange filled circles. Figure is not to scale.

**Table 1** Up-regulated (≥4×) genes in *A. salmonicida fur* null mutant compared to wild-type.

| VSAL_nr | gene | Annotation | Δ*fur*/wt | Fur-box[**] |
|---|---|---|---|---|
| *Siderophore biosynthesis and transport* | | | | |
| VSAL_I0134[*] | bibA | Bisucaberin siderophore biosynthesis protein A | 92.6 | x |
| VSAL_I0135 | bibB | Bisucaberin siderophore biosynthesis protein B | 48.2 | x |
| VSAL_I0136 | bibC | Bisucaberin siderophore biosynthesis protein C | 11.1 | x |
| VSAL_I0137 | bitA | TonB-dependent iron-siderophore receptor precursor | 9.3 | x |
| VSAL_II0148 | | 2Fe-2S binding protein, siderophore ferric reductase | 8.0 | x |
| VSAL_II0150 | fhuC | ferrichrome transport ATP-binding protein FhuC | 7.0 | x |
| VSAL_II0151 | fhuD | ferrichrome-binding periplasmic protein | 12.5 | x |
| VSAL_II0152 | fhuB | ferrichrome transport protein FhuB | 6.7 | x |
| VSAL_II0907 | | iron(III) ABC transporter, periplasmic iron-compound-binding (pseudo) | 5.9 | x |
| VSAL_II0908 | hatC | iron(III) ABC transporter, ATP-binding protein | 11.2 | x |
| VSAL_II0909 | desA | ferrioxamine B receptor | 18.8 | x |
| *TonB systems* | | | | |
| VSAL_I1751 | tonB1 | TonB protein (pseudogene) | 18.8 | x |
| VSAL_I1752 | exbB1 | TonB system transport protein ExbB1 | 25.2 | x |
| VSAL_I1753 | exbD1 | TonB system transport protein ExbD1 | 28.4 | x |
| VSAL_II0110 | | TonB dependent receptor | 55.8 | x |
| VSAL_II0111 | | putative exported protein | 35.3 | x |
| VSAL_II0112 | tolR2 | biopolymer transport protein TolR | 25.7 | x |
| VSAL_II0113 | exbB2 | TonB system transport protein ExbB2 | 17.3 | x |
| VSAL_II0114 | exbD2 | TonB system transport protein ExbD2 | 27.6 | x |
| VSAL_II0115 | tonB2 | TonB protein | 30.1 | x |
| VSAL_II0116 | | putative exported protein | 23.4 | x |
| *Heme uptake and utilization* | | | | |
| VSAL_I1734 | | heme receptor (pseudogene) | 6.6 | x |
| VSAL_I1749 | huvX | heme uptake and utilization protein HuvX | 20.2 | x |
| VSAL_I1750 | phuW | putative coproporphyrinogen oxidase PhuW | 39.7 | x |
| VSAL_I1754 | huvB | heme transporter protein HuvB, periplasmic binding protein | 39.7 | x |
| VSAL_I1755 | huvC | heme transporter protein HuvC, transmembrane permease component | 13.5 | x |
| VSAL_I1756 | huvD | heme transporter protein HuvD, ATP-binding component | 5.8 | x |
| *small RNA* | | | | |
| VSAL_I3102s | ryhB | small RNA RyhB | 43.7 | x |
| VSAL_II2005s | | VSAsRNA006 | 4.0 | |
| *Other transport* | | | | |
| VSAL_I1819 | | outer membrane protein A | 5.9 | |
| VSAL_I2067 | zntA | lead, cadmium, zinc and mercury-transporting ATPase | 8.5 | |
| VSAL_I2891 | vcmD | multidrug efflux pump | 8.5 | x |
| VSAL_II0118 | | membrane protein | 16.9 | |

**Table 1** (*continued*)

| VSAL_nr | gene | Annotation | Δ*fur*/wt | Fur-box[**] |
|---|---|---|---|---|
| VSAL_II0119 | | putative exported protein | 25.7 | |
| VSAL_II0120 | | nickel transporter | 16.7 | |
| VSAL_II0121 | | putative exported protein | 16.7 | |
| VSAL_II0122 | | putative membrane protein | 8.7 | |
| VSAL_II0123 | | zinc ABC transporter periplasmic substrate binding protein | 7.4 | |
| VSAL_II0124 | | zinc ABC transporter ATP binding protein | 6.3 | |
| VSAL_II0125 | | zinc ABC transporter permease | 4.1 | |
| VSAL_II0149 | | MFS transporter | 5.6 | |
| VSAL_II1043 | | cation efflux pump, cobalt-zinc-cadmium resistance protein | 5.7 | |
| VSAL_II1067 | potE | putrescine-ornithine antiporter | 5.0 | |
| *Metabolism* | | | | |
| VSAL_I1785 | | thiol oxioreductase | 5.7 | |
| VSAL_I1786 | | peptidase, putative iron-regulated | 8.2 | x |
| VSAL_I2892 | | methyltransferase | 12.4 | x |
| VSAL_II0932 | bcsA | cellulose synthase catalytic subunit | 6.1 | |
| VSAL_II1066 | speF | ornithine decarboxylase, inducible | 7.4 | |
| *Cell envelope* | | | | |
| VSAL_I1328 | | putative membrane associated peptidase | 4.4 | |
| VSAL_I1783 | | putative lipoprotein | 4.4 | |
| VSAL_I1784 | | putative lipoprotein | 5.0 | |
| VSAL_I1820 | | putative lipoprotein | 4.0 | |
| VSAL_I1864 | | putative membrane protein | 20.1 | x |
| VSAL_II0074 | | membrane protein | 67.3 | x |
| VSAL_II0868 | | putative lipoprotein | 8.0 | x |
| VSAL_II0931 | | membrane protein (fragment) | 4.8 | |
| VSAL_II0933 | | putative exported protein | 6.2 | |
| VSAL_II0937 | | membrane protein | 4.0 | |
| *Unknown function* | | | | |
| VSAL_I0881 | | putative exported protein | 15.7 | x |
| VSAL_I0882 | | putative exported protein | 14.1 | x |
| VSAL_I0883 | | putative exported protein | 14.4 | x |
| VSAL_I0884 | | putative exported protein | 5.0 | x |
| VSAL_II0469 | | hypothetical protein | 4.5 | |
| VSAL_II0934 | | hypothetical protein | 4.0 | |

**Notes.**

[*]*p*-value not analyzed.

[**]fur-box predictions from *Ahmad et al. (2009)*.

were curious to see if *as*Fur-related genes are found clustered at specific regions of Chr I, perhaps with relevance to their expression levels due to gene dosage.

In our experimental setup the average RPKM value for the upper half of Chr I (i.e., the region closest to *ori* C) is significantly higher compared to that of the lower half (660/330 for wild-type and 560/397 for *fur* null mutant). Gene dosage effects have yet to be demonstrated for Chr II (*Dryselius et al., 2008*; *Toffano-Nioche et al., 2012*), which is in agreement with the RPKM values in our experiment (RPKM values are similar for the upper and lower halves of the chromosome). Differentially expressed genes appear to be relatively evenly distributed on the chromosome, except for some clustering of genes between Chr I pos. 1.85–2.01 Mb. They represent a TonB1 system, heme transport and utilization, and cell envelope genes (up-regulated genes), and oxidative stress response, metabolism and sRNAs (down-regulated genes). In other words, there is apparently no clear pattern with respect to *as*Fur-regulated genes and their genomic position. It is interesting to note, however, that the bisucaberin biosynthesis gene cluster and *ryhB* (encodes the RyhB sRNA) are both located close to *oriC*. We have previously reported that the bisucaberin biosynthesis system is included in the immediate response to iron limitations in *A. salmonicida* (*Thode et al., 2015*), and its genomic location may contribute to the high level of expression and fast response to iron starvation.

## asFur regulates iron acquisition systems

As expected, a high proportion of up-regulated genes (28 of 64) are directly associated with iron metabolism, e.g., siderophore biosynthesis and transport, TonB systems (delivery of energy to iron transport), and heme uptake and utilization. The most up-regulated (92×) gene is *bibA*, which together with the two downstream genes *bibBC* (48× and 11× up-regulated in the *fur* null mutant, respectively) are responsible for producing the siderophore bisucaberin. The overall transcription level for the *bibABC* genes also varies dramatically (see Fig. 3A), and follows a trend that more reads map to the first genes of the operons. Therefore, the expression pattern follows the differential expression values for the operon (i.e., 92×, 48× and 11×). Interestingly, within the large *Vibrionaceae* family *bibABC* are restricted to *A. salmonicida* and *Aliivibrio logei* (*Kadi, Song & Challis, 2008*; *Thode et al., 2015*), and are in *A. salmonicida* (together with a siderophore transport system, *bitABCDE*) flanked by transposable elements (i.e., a genomic island; see (*Hjerde et al., 2008*)). Homology search with the BibABC amino acid sequences from *A. salmonicida*, revealed that the close relative *Aliivibrio wodanis* also possesses the bisucaberin biosynthesis system. The coverage and identity percentage from blastP (with *A. salmonicida* sequences used as query) were 87% identity over 100% coverage for BibA, 90% identity over 99% coverage for BibB and 89% identity over 100% coverage for BibC.

Other siderophore receptors and iron-related transport systems that are significantly up-regulated in the *fur* null mutant include the ferrichrome transport system [VSAL_II0150–0152 (6.7–12.5×)], the ferrioxamine B receptor [VSAL_II0909 (18.8×)] and its associated ABC transporters [VSAL_II0907 (5.9×) and II0908 (18.8×)]. A siderophore ferric reductase [VSAL_II0148 (8×)] responsible for removing iron from the siderophore, the TonB1 system [VSAL_I1751–1753 (18.8–28.4×)], and finally *huvB*, *huvC* and *huvD*

**Table 2  Down-regulated ($\leq -3\times$) genes in *A. salmonicida fur* null mutant compared to wild-type.**

| VSAL_nr | gene | annotation | $\Delta fur$/wt | sRNA target |
|---|---|---|---|---|
| *Motility/ chemotaxis* | | | | |
| VSAL_I0799 | | methyl-accepting chemotaxis protein | −3.5 | |
| VSAL_I2193[*] | | methyl-accepting chemotaxis protein | −3.6 | |
| VSAL_I2317 | *flaE* | flaggelin subunit E | −5.1 | |
| VSAL_I2318 | *flaD* | flaggelin subunit D | −4.3 | |
| VSAL_I2319 | *flaC* | flaggelin subunit C | −6.2 | |
| VSAL_I2517 | *flaF* | flaggelin subunit F | −3.9 | |
| VSAL_I2771 | *motX* | sodium-type polar flagellar protein MotX | −5.0 | |
| *Oxidative stress response* | | | | |
| VSAL_I1858 | *sodB* | superoxide dismutase [Fe] | −3.1 | RyhB |
| VSAL_II0215 | *catA* | catalase | −3.4 | |
| *Metabolism* | | | | |
| VSAL_I0122 | *prlC* | oligopeptidase A | −3.2 | |
| VSAL_I0421 | *cysN* | sulfate adenylyltransferase subunit 1 | −3.4 | RyhB |
| VSAL_I0422 | | ion transporter superfamily protein | −3.8 | RyhB |
| VSAL_I0423 | *cysC* | adenylylsulfate kinase | −4.0 | |
| VSAL_I1133 | *hisG* | ATP phosphoribosyltransferase | −3.4 | |
| VSAL_I1769 | *nrdA* | ribonucleoside-diphosphate reductase 1 alpha chain | −3.8 | |
| VSAL_I1857 | *queD* | queuosine biosynthesis protein | −4.0 | |
| VSAL_II0666 | *idnK* | thermosensitive gluconokinase | −4.4 | |
| VSAL_II0846 | | putative acetyltransferase | −3.4 | |
| VSAL_II1026 | | putative tryptophanyl-tRNA synthetase | −6.4 | RyhB |
| *small RNA* | | | | |
| VSAL_I4000s | | VSsRNA001 | −4.1 | |
| VSAL_I4069s | | VSsRNA070 | −3.4 | |
| VSAL_I4100s | | VSsRNA 101 | −4.1 | |
| VSAL_I4139s | | VSsRNA140 | −3.9 | |
| *Chaperones/heat shock proteins* | | | | |
| VSAL_I0017 | *groL1* | 60 kda chaperonin 1 | −3.2 | |
| VSAL_I0018 | *groS1* | 10 kDa chaperonin 1 | −3.9 | |
| VSAL_I0814 | *htpG* | chaperone protein HtpG (heat shock protein HtpG) | −3.2 | |
| *Cell envelope/ transport* | | | | |
| VSAL_I1813 | *tcyP* | L-cystine transporter | −8.6 | RyhB, VSAL_II2005s |
| VSAL_II0853 | | MFS transporter | −4.0 | |
| VSAL_II0854 | | secretion protein, HlyD family | −3.9 | |
| VSAL_II1062 | | membrane protein | −3.3 | |
| *Unknown function* | | | | |
| VSAL_I0424 | | hypothetical protein | −3.2 | RyhB |
| VSAL_I2064 | | conserved hypothetical protein | −4.0 | |
| VSAL_II0168 | | putative exported protein | −7.9 | |
| *Mutated gene/control gene* | | | | |
| VSAL_I0833 | *fur* | ferric uptake regulator protein | −128.7 | RyhB |

**Notes.**

*fur-box predicted in *Ahmad et al. (2009)*.

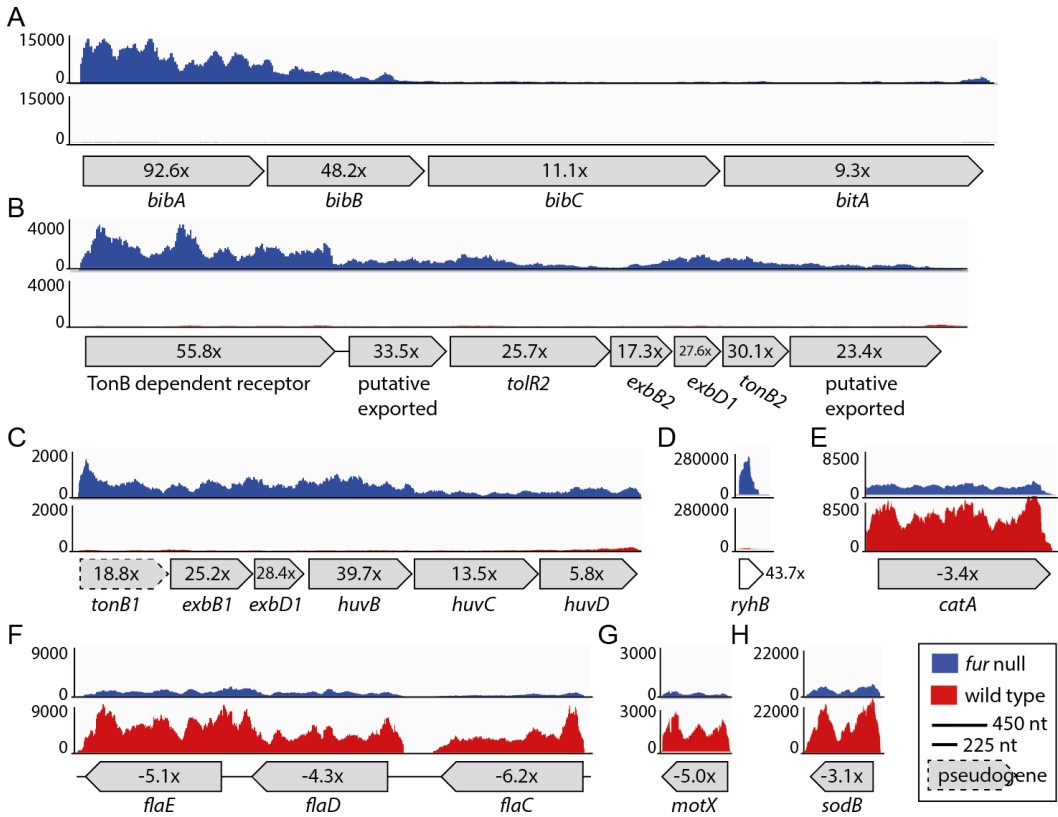

**Figure 3  Relative expression levels for a selection of CDSs.** (A) VSAL_I0134–VSAL_I0137; (B) VSAL_II0110–VSAL_II0116; (C) VSAL_I1751–VSAL_1756; (D) VSAL_I3102s; (E) VSAL_II0215; (F) VSAL_I2317–VSAL_I2319; (G) VSAL_I2771; (H) VSAL_I1858. $Y$-axis indicate the number of mapped reads. Red and blue curves represent mapped reads for wild-type and *fur* null mutant, respectively. The synteny of CDSs are shown below the graphs with associated numbers indicating the differential expression value ($\Delta fur$/wt).

[VSAL_I1754–I1756 (5.8 −39.7×)] responsible for heme transport, are up-regulated in the *fur* null mutant (see Fig. 3C for expression details). The heme uptake and utilization gene *huvX* [VSAL_I1749 (20.2×)] and *phuW* [VSAL_I1750 (39.7×)], which encode a putative coproporphyrinogen oxidase believed to be responsible for removing iron from heme, are highly up-regulated in the *fur* null mutant. The TonB2 system [VSAL_II0110–II0116 (55.8–17.3× up-regulated)] (Fig. 3B), iron(III) ABC transporters [VSAL_II0907 (5.9×) and II0908 (11.2×)] and a siderophore receptor gene *desA* [VSAL_II0909 (18.8×)] are all highly up-regulated. Interestingly, *feoABC* (VSAL_I2257–I2259) that encode the ferrous iron transport system, are apparently not strongly regulated by Fur, as only *feoC* from this system has an up-regulation ≥2× (i.e., 2.3×).

In summary, removal of the *fur* gene results in up-regulation of 28 genes directly associated with iron homeostasis (siderophore biosynthesis, transport and utilization, heme transport and utilization, ABC transporters and TonB1 and TonB2 systems). *bibA* is by far the most up-regulated (92×) gene, whereas the remaining iron-relevant genes are up-regulated 55–5×.

### asFur regulates several metal transport systems

As shown in Fig. 1 and Table 1, several transport systems are up-regulated in the *fur* null mutant. *as*Fur may be involved in the homeostasis of other metals than iron, as multi metal resistance protein genes, a multidrug efflux pump, and nickel and zinc transporter genes are up-regulated. In detail; the multi metal resistance genes *zntA* (VSAL_I2067) and VSAL_II0143 are up-regulated 8.5× and 5.7×, respectively. The multidrug efflux pump encoded by *vcmD* (VSAL_ I2891) is 8.5× up-regulated. A large operon (VSAL_II0118-II0125) with annotated nickel and zinc transporters is also up-regulated 4.1–25.7×. Also, the outer membrane protein A gene (VSAL_I1819), a MFS transporter gene (VSAL_II0149) and *potE* (VSAL_II1067) are up-regulated 5.9×, 5.6× and 5.0×, respectively.

### Down-regulated genes in asFur null mutant

Fur primarily functions as a repressor. The down-regulated genes in our study (i.e., in the *fur* null mutant) are expected to be positively regulated by *as*Fur in the wild-type, either via the repression of *ryhB* (or other sRNAs with similar function), which typically destabilizes its mRNA targets (*Oglesby-Sherrouse & Murphy, 2013*), or by direct stimulation of expression by *as*Fur itself. In this study, we cannot conclusively distinguish between these two possibilities, although we have predicted potential targets of RyhB and other up-regulated sRNAs (see below).

Table 2 shows 34 down-regulated genes in the *fur* null mutant compared to wild-type. Overall, the Δ*fur*/wt values for down-regulated genes are significantly lower than that of up-regulated genes (the strongest down-regulation is −8.6×, when excluding *fur* that has been deleted from the genome). In Table 2 we therefore present genes that are ≤ − 3× down-regulated. The majority of the genes are categorized as "motility/chemotaxis" or "metabolism". "Metabolism" genes are involved in different pathways such as amino acid, energy, nucleotide, carbon etc. Moreover, several motility and chemotaxis genes are down-regulated between −3.5× and −6.3×. Of these, four encode flagellin subunits [*flaC-flaE* (VSAL_I2317- I2319) (Fig. 3) and *flaF* VSAL_I2517)], one encodes a sodium-type polar flagellar protein MotX (VSAL_2771) (Fig. 3), and two encode methyl-accepting chemotaxis proteins (VSAL_I0799 and VSAL_I2193). Three heat shock proteins encoded by *groL1* (VSAL_I0017), *groS1* (VSAL_I0018) and *htpG* (VSAL_I0814) are also down-regulated. Heat shock proteins are involved in protein folding and unfolding, cell cycle control, transport and stress responses amongst others. Transcriptome studies of a Δ*fur* mutant in *V. vulnificus* have also shown a down-regulation of heat shock protein genes, chemotaxis protein genes and motility-associated genes (*Pajuelo et al., 2016*). Two oxidative stress response protein encoding genes, *sodB* and *catA* (VSAL_I1858 and VSAL_II0215), are down-regulated (Fig. 3). SodB is an iron binding protein and a RyhB target in other organisms, and CatA is a heme-binding protein.

In summary, differentially down-regulated genes in the *A. salmonicida fur* null mutant have significantly lower differential expression values than the up-regulated genes possibly due to, in part, secondary regulatory effects rather than direct regulation by Fur. The majority of down-regulated genes have functions in chemotaxis, motility, heat shock and oxidative stress response.

## Identification of sRNAs with roles in iron homeostasis

ncRNAs represent an important part of regulons in bacteria, often controlling critical and early steps in pathways (*Gottesman, 2005*). We therefore set out to explore the presence and function of sRNAs in our RNA-seq dataset. Table 1 and Fig. 3D already showed us that *ryhB* is up-regulated 43× in the *fur* null mutant, which strongly supports that RyhB in *A. salmonicida* has a similar role in iron homeostasis as what was established for its homologs in e.g., *E. coli* (*Masse, Vanderpool & Gottesman, 2005*; *Seo et al., 2014*) and *V. cholerae* (*Davis et al., 2005*). Here, RyhB is produced under low-iron conditions and stops production of iron-using and iron-storing proteins, and therefore contributes to a lowered demand for iron (*Jacques et al., 2006*; *Smaldone et al., 2012*).

To search for other sRNAs with potential roles in iron homeostasis we re-analyzed the RNA-seq dataset. The rational was that any Fur-regulated sRNA gene are likely candidates to have roles in iron metabolism by targeting specific mRNAs for degradation. One sRNA gene (VSAL_II2005s) that fulfilled this criterion was identified among 252 sRNA genes that we predicted in a previous work (*Ahmad et al., 2012*). VSAL_II2005s was up-regulated 4×. Furthermore, we analyzed the RNA-seq data using Rockhopper. Rockhopper predicts ncRNAs from RNA-seq data. The sRNAs predicted by Rockhopper were manually curated using the Artemis software. Briefly, to be accepted as a true sRNA, its gene had to be (i) located in an intergenic region, (ii) between 30–350 nt in length, (iii) located 30 nt or more from the nearest CDS if on the same strand, and 10 nt if on the complementary strand.

Ninety-three potential sRNA were predicted using Rockhopper. Seventeen were kept after manual curation, eight of which overlapped or located on the complementary strand of previously predicted sRNAs (*Ahmad et al., 2012*). These eight sRNAs are VSAL_I4057s, VSAL_I4069s and VSAL_I4164s (overlapping), and VSAL_I4107s, VSAL_I4164s, VSAL_I4189s, VSAL_II2008s and VSAL_II2050s (complementary). Of the remaining nine new sRNAs identified by Rockhopper and manual curation, six are located on Chr I and three on Chr II (see Fig. 4). sRNAs 4 and 7 both contain sORFs, which potentially encode small proteins (see Material and methods) (*Hobbs et al., 2011*). In general, reads that map to the region predicted by Rockhopper seem to be a sRNA gene. However, for sRNA 8 reads map to a larger region surrounding the region predicted by Rockhopper (see Fig. 4H). This discrepancy is likely due to that the sRNA is longer than predicted, or alternatively a false positive. The nine new sRNAs were added to the *A. salmonicida* genome annotation using Artemis, and the RNA-seq data was re-analyzed for differentially expressed genes using EDGE-pro and DESeq. Two of the sRNAs, i.e., number 1 and 9, were up-regulated 2.2× and 2.5× in the *fur* null mutant, respectively. Homology searches in ENA did not produce significant hits.

In summary, RyhB and a previously predicted sRNA (VSAL_II2005s) were up-regulated in the *A. salmonicida fur* null mutant. Nine new sRNAs were identified using Rockhopper and manual curation, of which two were differentially expressed (i.e., Figs. 4A and 4I). Notably, these newly identified sRNAs should be considered as putative until further evidence firmly establishes their presence, e.g., by Northern blot and RACE analyses.

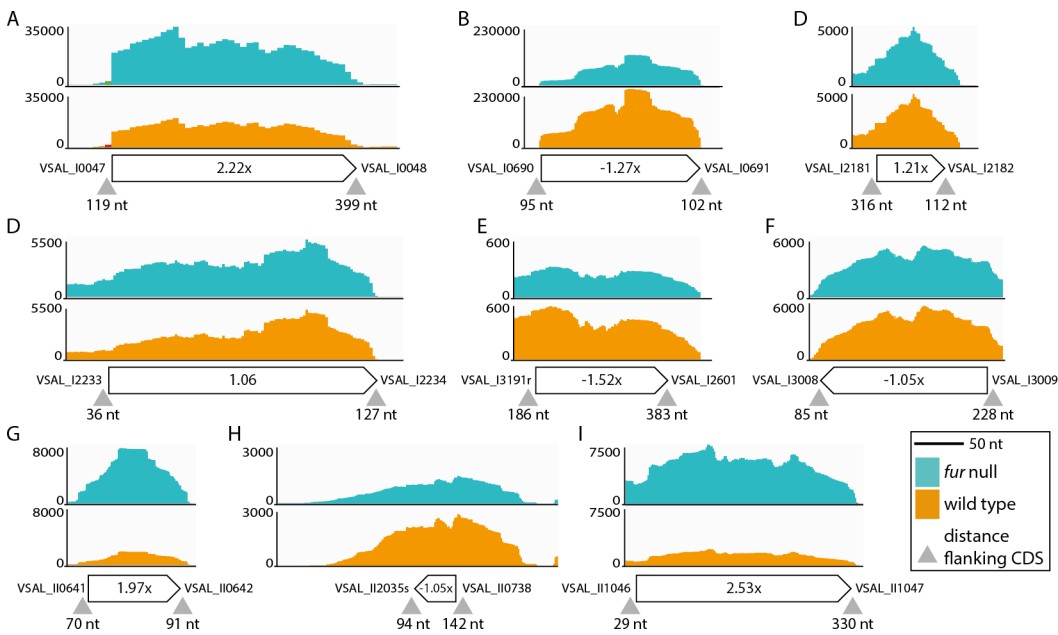

**Figure 4   sRNAs identified by Rockhopper and manual curation.** (A) sRNA 1 chromosome I position: 51134..51393. (B) sRNA 2 chromosome I position: 776673..776837. (C) sRNA 3 chromosome I position: 2343220..2343291. (D) sRNA 4 chromosome I position: 2405357..2405638. (E) sRNA 5 chromosome I position: 2812966..2813103. (F) sRNA 6 chromosome I position: 3259173..3259344. (G) sRNA 7 chromosome II position: 692443..692539. (H) sRNA 8 chromosome II position: 814013..814056. (I) sRNA 9 chromosome II position: 1141984..1142209. $Y$-axis indicate the number of mapped reads. Orange and turquoise curves represent mapped reads for wild-type and *fur* null mutant, respectively. sRNA genes are shown below curves, and associated numbers indicate the differential expression value ($\Delta fur$/wt). Small grey arrow heads indicate the distance in nt to flanking CDSs

## sRNA target predictions

Next, we used the TargetRNA2 and IntaRNA software to test if the up-regulated sRNAs identified above can explain some of the down-regulated protein-coding genes. The up-regulated sRNAs *ryhB*, VSAL_II2005s and new sRNAs 1 and 9 (see Figs. 4A and 4I) were tested for target binding towards the 34 down-regulated genes presented in Table 2. *ryhB* is up-regulated 43.7×, and typically targets mRNA for iron using and iron storage proteins (*Davis et al., 2005*; *Masse, Vanderpool & Gottesman, 2005*; *Mey, Craig & Payne, 2005*; *Murphy & Payne, 2007*; *Oglesby-Sherrouse & Murphy, 2013*). We expected to find same/similar targets in our dataset. Our results show that RyhB targets seven of the mRNAs listed in Table 2. *sodB* and *fur* represent known targets from other organisms (*Davis et al., 2005*; *Masse, Vanderpool & Gottesman, 2005*; *Mey, Craig & Payne, 2005*). The other identified targets are *cysN* (VSAL_I0421), VSAL_I0422, *tcyP* (VSAL_I1813), VSAL_II1026 and VSAL_I0424. Furthermore, we tested other known targets for complementarity to RyhB. Matches were found to *gltB* and *sdhC*, which were down-regulated 2.1× and 1.3×, respectively. We therefore consider *gltB* as a potential RyhB target in *A. salmonicida*, whereas *sdhC* is probably not (due to the weak regulation). In *E. coli* K-12 and *Bacillus Subtilis*, GltB is an iron-sulfur binding protein (*Miller & Stadtman, 1972*; *Smaldone et*

al., 2012). Thus, down-regulation of *gltB* is an iron sparing strategy (*Jacques et al., 2006*; *Smaldone et al., 2012*).

Our target predictions for VSAL_II2005s (which was 4× up-regulated) suggest significant complementarity to *tcyP* (VSAL_I1813). Interestingly, *tcyP* was also identified as a RyhB target, which may explain why *tcyP* has a relative strong down-regulation of -8.6 × when compared to the other down-regulated genes. No potential targets were identified for sRNAs 1 and 9 in Fig. 4.

In summary, *as*RyhB appears to have similar regulatory functions as its known homologs from other model organisms, and may account for the down-regulation of seven of the 34 genes in Table 2. We also identified *tcyP* as a potential target for both RyhB and VSAL_II2005s. No complementarity was found between the newly identified sRNAs 1 and 9 and mRNAs corresponding to the down-regulated genes listed in Table 2.

## CONCLUDING REMARKS

We have studied the Fur regulon of *A. salmonicida* using gene knock out technology, and compared the transcriptome of the *fur* null mutant with its isogenic wild-type using RNA sequencing. Our results show that *as*Fur acts as a master regulator in *A. salmonicida* affecting ~7% of the CDSs, when threshold values were set to 2× differential expression and *p*-value ≤0.05. We also demonstrate that *as*Fur acts mainly as a repressor. This conclusion is based on that Δ*fur*/wt differential expression values of up-regulated genes in the *fur* null mutant are significantly higher than that of down-regulated genes. Furthermore, we demonstrated a strong *gene dosage effect* for Chr I. This result adds to the growing list of *Vibrionaceae* bacteria where the transcription level is, statistically, highest for chromosomal regions surrounding $oriC_I$, and weaker for genes located on the opposite end of the chromosome (surrounding $terC_I$). Finally, we identify sRNAs with potential roles in iron homeostasis. The role for RyhB is well established, and in addition, we identified VSAL_II2005s, which was 4× up-regulated in a *fur* null mutant, and contains extensive potential for base pairing to the RyhB target *tcyP* (VSAL_I1813).

Our current data is in good overall agreement with our previous work (*Ahmad et al., 2012*; *Ahmad et al., 2009*; *Pedersen et al., 2010*; *Thode et al., 2015*). For example, our current data overlap with results from our previous works where *A. salmonicida* was subjected to low-iron conditions and global changes in gene expression was monitored using microarray (*Thode et al., 2015*). Twenty-eight of the 32 genes identified by microarray were ≥2× up-regulated in the *fur* null mutant. With the latest data we conclude that we today have a more accurate and fine-grained global understanding of the Fur regulon in *A. salmonicida*.

**Abbreviations**

| | |
|---|---|
| **ABC transporter** | ATP-binding cassette |
| **Fur** | Ferric Uptake Regulator |
| **ec**Fur | *Escherichia coli* Fur |
| **as**Fur | *Aliivibrio salmonicida* Fur |
| **sRNA** | small regulatory RNA |

| | |
|---|---|
| **ORF** | Open reading frame |
| **mRNA** | messenger RNA |
| **TCA** | tricarboxylic acid |
| **DNA** | Deoxyribonucleic acid |
| **RNA** | Ribonucleic acid |
| **bp** | base pair |
| **nt** | nucleotide |
| **LB** | Luria Bertani broth/Lysogen Broth |
| **tRNA** | transfer RNA |
| **rRNA** | ribosomal RNA |
| **Chr** | Chromosome |
| **MFS transporter** | major facilitator superfamily transporter |
| **h** | hours |
| **PCR** | Polymerase Chain Reaction |
| **OD** | optical density |
| **wt** | wild-type |
| **RPKM** | reads per kilo base per million mapped reads |
| **RNA-seq** | RNA sequencing |
| **rpm** | rounds per minute |
| *AS* | *Aliivibrio salmonicida* |
| **sORF** | small open reading frame |
| **ncRNA** | non-coding RNA |
| Δ*fur* | *fur* null mutant. |

## ACKNOWLEDGEMENTS

The sequencing service was provided by the Norwegian Sequencing Centre (www.sequencing.uio.no), a national technology platform hosted by the University of Oslo and supported by the "Functional Genomics" and "Infrastructure" programs of the Research Council of Norway and the Southeastern Regional Health Authorities.

### Funding

This work was funded by the Norwegian National Graduate School in Structural Biology (Biostruct), and UiT the Arctic University of Norway. The funders had no role in study design, data collection and analysis, decision to publish, or preparation of the manuscript.

### Grant Disclosures

The following grant information was disclosed by the authors:
Norwegian National Graduate School in Structural Biology (Biostruct).
UiT The Arctic University of Norway.

### Competing Interests

The authors declare there are no competing interests.

## Author Contributions

- Sunniva Katharina Thode conceived and designed the experiments, performed the experiments, analyzed the data, contributed reagents/materials/analysis tools, wrote the paper, prepared figures and/or tables, reviewed drafts of the paper.
- Cecilie Bækkedal analyzed the data, contributed reagents/materials/analysis tools, wrote the paper, prepared figures and/or tables, reviewed drafts of the paper.
- Jenny Johansson Söderberg performed the experiments, reviewed drafts of the paper.
- Erik Hjerde analyzed the data, contributed reagents/materials/analysis tools, reviewed drafts of the paper.
- Hilde Hansen conceived and designed the experiments, contributed reagents/materials/analysis tools, reviewed drafts of the paper.
- Peik Haugen conceived and designed the experiments, contributed reagents/materials/analysis tools, wrote the paper, prepared figures and/or tables, reviewed drafts of the paper.

## Data Availability

RNA sequencing data are accessible in the European Nucleotide Archive (ENA) under accession number PRJEB17700 (available from 7th of January 2016).

## Supplemental Information

Supplemental information for this article can be found online at http://dx.doi.org/10.7717/peerj.3461#supplemental-information.

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
