# Peer review of "Construction of a fur null mutant and RNA-sequencing provide deeper global understanding of the Aliivibrio salmonicida Fur regulon"

_PeerJ, doi:10.7717/peerj.3461_

## Round 0.1 · original submission · Major Revisions

· Academic Editor

Major Revisions

I agree fully with reviewer 2 that your paper is generally well-written, your assays seem well designed to me and your results are indeed interesting. Nevertheless, as you can see from both reviews, there are a few concerns with respect to the validity of the findings, one reviewer recommends for the RNA-Seq analysis a statistical test for multiple comparisons, the second comments on in his their contradictory predictions around siderophore and heme uptake proteins. Please comment on both concerns either following the suggestions of the reviewers with respect to additional exps/analysis or preparing hard arguments why you you would not follow their advice. Additionally, especially reviewer 1 made very detailed comments on the basic findings and the experimental design, thus I would also advise you to consider to include additional details regarding the conjugation experiments used to construct the fur mutant, as the mentioned diagram of the constructs or an agarose gel of the PCR for WT and fur mutant, and to go carefully through the very detailed remarks/concerns/suggestion in both reviews and address them in your rebuttal letter. I am looking forward to see your revised article.

·

Basic reporting

1. There are a few typos throughout. Lines 56 and 351 are examples.

2. On line 437, the authors mention iron-sparing, but there is an absence of appropriate references. Also, lines 392-394 imply the iron-sparing concept. I would suggest adding the following publications:

Jacques, J.-F., Jang, S., Prévost, K., Desnoyers, G., Desmarais, M., Imlay, J. and Massé, E. (2006), RyhB small RNA modulates the free intracellular iron pool and is essential for normal growth during iron limitation in Escherichia coli. Molecular Microbiology, 62: 1181–1190.

Gaballa A., Antelmann H., Aguilar C., Khakh S. K., Song K. B., Smaldone G. T., et al. The Bacillus subtilis iron-sparing response is mediated by a Fur-regulated small RNA and three small, basic proteins. (2008). Proc. Natl. Acad. Sci. U.S.A. 105, 11927–11932

Smaldone GT, Revelles O, Gaballa A, Sauer U, Antelmann H, Helmann JD. A Global Investigation of the Bacillus subtilis Iron-Sparing Response Identifies Major Changes in Metabolism. (2012) Journal of Bacteriology 194:2594-2605.

Seo SW, Kim D, Latif H, O’Brien EJ, Szubin R, Palsson BO. Deciphering Fur transcriptional regulatory network highlights its complex role beyond iron metabolism in Escherichia coli. (2014) Nature Communications 5:4910.

3. Figure 2 is visually appealing, but the insertion of wt and fur transcript levels doesn't allow for simple assessment of the changes in global gene expression. I suggest focusing on pertinent Fur-regulated genes (bib operon, etc) and generate additional figures similar to Fig2B in the following publication:

Srikumar S, Kröger C, Hébrard M, Colgan A, Owen SV, Sivasankaran SK, et al. (2015) RNA-seq Brings New Insights to the Intra-Macrophage Transcriptome of Salmonella Typhimurium. PLoS Pathog 11(11): e1005262. doi:10.1371/journal. ppat.1005262.

4. The authors used a cutoff of >2 or <-2-fold change in gene expression for further analysis. However, Table 1 and 2 only show genes expressed >4 and <-3-fold, respectively. This data set appears to be incomplete. Could the authors provide rationale for the discrepancies in fold changes? Also, please provide the complete list of significantly regulated genes.

5. Data in Table 3 may be better shown as a figure that includes the above suggestion in #3. Displaying the traits of each sRNA, as listed in lines 402-404, would be informative.

6. Thank you for providing the RAW data for RNA Seq for analysis. Within the text, it would be informative to tabulate or describe the details of each data file.Lines 161-173 mentions some of this data, but the % mapped to reference genome, total reads per sample, etc are not shown. These are typically included with RNA seq publications.

Experimental design

1. There is much emphasis on the contribution of Fur to virulence in pathogens. Although this work focuses on how Fur regulates gene expression in A. salmonicida, there are no experiments that test Fur's contribution to virulence. Without these experiments, there is still a knowledge gap regarding Fur and virulence within A. salmonicida.

2. Lines 124-130. Why weren't the growth conditions for experiments and RNA-Seq experiments the same? This causes great difficulty in elucidating Fur's contribution to gene regulation with the observations in Figs S1 and 2.

3. Please include additional details regarding the conjugation experiments used to construct the fur mutant. A diagram of the constructs could be informative. Also, an agarose gel of the PCR for WT and fur mutant would be beneficial.

4. Within Table 2, genes associated with motility were down regulated in the fur mutant. Were experiments performed to test differences in motility between WT and fur mutant?

5. For Fig S1, panel B is sufficient and most relevant to interpreting changes in growth kinetics. Panel A is not needed.

6. For FigS2, please use semi-log graphs for growth curves while keeping the same values on the y-axis.

7. The changes in catA and heme acquisition genes provide insight into the contribution of Fur to the observed phenotypes in Figs S1 and 2. Given the down regulation of catA, changes in heme acquisition genes, and sensitivity to H2O2, did the Authors test the role of adding exogenous heme to the growth medium?

8. The RNA-Seq analysis should include a statistical test for multiple comparisons, ie, FDR. It may be beneficial to reanalyze the data with an FDR for statistical cutoff. This is a common approach to analyzing RNA-Seq data.

9. One key caveat to this work is the lack of an independent method to confirm the RNA-Seq data. qRT-PCR or reporter assays would address this concern.

10. Additional data are needed to address the presence of newly identified sRNAs. This part appears incomplete.

Validity of the findings

1. The RNA-Seq data needs to be reanalyzed using a test for multiple comparisons.

Reviewer 2 ·

Basic reporting

I enjoy reading this article. This is a well-written article, the assays are well design and results are interesting. However, I have some concerns about this article. The results are based only on gene expression and do not consider posttranslational and posttranscriptional regulation. My concern is that the author assumed that A. salmonicida synthesized and secrete siderophores depending on Fur regulation or up-regulated Hem receptors or siderophore receptors in a Fur-dependent fashion. For instance, the feoABC system is an inner membrane system for ferrous iron, not for siderophore uptake system. Usually, pathogens have either, siderophore or Hem uptake system, it is unusual that they have both.

Since the results authors are speculative and contradictory, the authors need to show that A. salmonisida synthesizes siderophores in a Fur-dependent fashion. Also, the authors need to show that Fur mutants are able to uptake heme or the outer membrane proteins up-regulated are heme uptake proteins. All these assays are classic experiments in iron bacteriology.

Experimental design

The assays are well designed. No comments here.

Validity of the findings

As I mentioned previously, there are some contradictory predictions around siderophore and heme uptake proteins. This isn´t clear and needs to be addressed. Classic assays need to be performed.

Additional comments

I enjoy reading this article. This is a well-written article, the assays are well design and results are interesting. However, I have some concerns about this article. The results are based only on gene expression and do not consider posttranslational and posttranscriptional regulation. My concern is that the author assumed that A. salmonicida synthesized and secrete siderophores depending on Fur regulation or up-regulated Hem receptors or siderophore receptors in a Fur-dependent fashion. For instance, the feoABC system is an inner membrane system for ferrous iron, not for siderophore uptake system. Usually, pathogens have either, siderophore or Hem uptake system, it is unusual that they have both.

Good piece of work, just need some improvements in with the iron up-take concepts.

---

## Round 0.2 · Minor Revisions

· Academic Editor

Minor Revisions

Thanks a lot for the thorough revision, the manuscript is nearly ready for acceptance except for two very minor details that were raised by one reviewer and that I think should be addressed briefly:

- The Seo et al citation is present in the text, but is missing within the references section.
- Could you please identify which of the fur mutant lanes represent the isolate used in the current studies (referring to remark 3 in the experimental design section)?

I am looking forward to receive your revision as soon as possible and promise that once the two minor issues are addressed/fixed I will handle your paper without further delay.

·

Basic reporting

1. Thank you for addressing this item. The text has been improved.
2. I appreciate the authors addition of these references. Please note that the Seo et al citation is present in the text, but is missing within the references section.
3. Thank you for adding this figure to the manuscript.
4. Thank you for adding the complete data set to this work. It is well organized.
5. I appreciate the authors adding this figure to the manuscript.
6. Thank you for adding these important data to the current work.

Experimental design

1. Thank you for making these changes.
2. Thank you for this clarification.
3. Thank you for adding this to the current work. Could the authors please identify which of the fur mutant lanes represent the isolate used in the current studies?
4. Thank you for addressing this aspect of the role of Fur in motility.
5. Thank you for clarifying the rationale for including Fig S2A.
6. Thank you for providing the rationale for this figure.
7. Thank you for considering the role of heme on Fur’s impact on the cell in future experiments.
8. Thank you for addressing this concern.
9. The authors provided a logical argument for the exclusion of an additional method to verify RNA-Seq data. To clarify, this reviewer suggested the inclusion of any suitable method for verification. This could include either qRTPCR, operon fusions, or other methods.
10. Thank you for including these lines in the text.

Validity of the findings

1. Thank you for addressing this concern.

---

## Round 0.3 · accepted · Accept

· Academic Editor

Accept

Thank you for addressing the two minor remarks left over from the last review round. The remarks were addressed properly and the manuscript is now ready for publication.